# Bone Fractures Numerical Analysis in a Femur Affected by Osteogenesis Imperfecta

**DOI:** 10.3390/children8121177

**Published:** 2021-12-14

**Authors:** Viridiana Ramírez-Vela, Luis Antonio Aguilar-Pérez, Juan Carlos Paredes-Rojas, Juan Alejandro Flores-Campos, Fernando ELi Ortiz-Hernández, Christopher René Torres-SanMiguel

**Affiliations:** 1Instituto Politécnico Nacional, Escuela Superior de Ingeniería Mecánica y Eléctrica Unidad Zacatenco, Sección de Estudios de Posgrado e Investigación, Ciudad de Mexico 07738, Mexico; vramirezv0700@alumno.ipn.mx (V.R.-V.); laguilarp@ipn.mx (L.A.A.-P.); 2Instituto Politécnico Nacional, Centro Mexicano para la Producción más Limpia, Ciudad de Mexico 07340, Mexico; jparedes@ipn.mx; 3Instituto Politécnico Nacional, Unidad Profesional Interdisciplinaria en Ingeniería y Tecnologías Avanzadas, Ciudad de Mexico 07340, Mexico; jaflores@ipn.mx; 4Instituto Politécnico Nacional, Escuela Superior de Ingeniería Mecánica y Eléctrica, Unidad Culhuacán, Ciudad de Mexico 04260, Mexico; fortizh@ipn.mx

**Keywords:** biomechanics, bone fractures, osteogenesis imperfecta

## Abstract

This work presents a non-invasive methodology to obtain a three-dimensional femur model of three-year-old infants affected with Osteogenesis Imperfecta (OI) type III. DICOM^®^ Files of a femur were processed to obtain a finite element model to assess the transverse, the oblique, and the comminuted fractures. The model is evaluated under a normal walking cycle. The loads applied were considered the most critical force generated on the normal walking cycle, and the analyses considered anisotropic bone conditions. The outcome shows stress concentration areas in the central zone of the diaphysis of the femur, and the highest levels of stress occur in the case of the comminuted fracture, while the transverse fracture presents the lowest values. Thus, the method can be helpful for determining the bone fracture behavior of certain pathologies, such as osteogenesis imperfecta, osteopenia, and osteoporosis.

## 1. Introduction

The Osteogenesis Imperfecta (OI) covers a set of diseases mainly characterized by a heterogeneous connective tissue disorder correlated to collagen production. Its prevalence is estimated at 1 in 10,000–15,000 children [1]. People related to these disorders commonly present abnormal bone structures due to the increased rate of collagen production. Besides this effect, they exhibit folding and intracellular transport modifications and difficulties of incorporation in different degrees [2]. Therefore, children with OI have low bone density associated with incremental bone fragility [3].

The management of patients with OI is multidisciplinary. It requires different specialists, such as geneticists, pediatric endocrinologists, traumatologists, rehabilitators, physiotherapists, otorhinolaryngologists, neurosurgeons, psychologists, and even engineers. All the treatments currently used to counteract the effects of OI can be classified into three groups: non-surgical, surgical, and medical [4]. Within the surgical group, procedures are carried out to correct bone deformities, where the planned fracture of the bone is carried out to align it. According to medical management, a telescopic nail counteracts the incidence of fractures and supports the bone [5,6,7,8].

OI children’s life is full of events that cause a fracture on their bones. The fracture usually happens with minimal or null trauma, but the injury could be equivalent to a fall from a height of one foot or less [9]. The OI affected exhibit fractures in the hip, clavicle, vertebrae, ribs, upper limbs, and lower limbs, the last of which being the area with the highest incidence of fracture, especially in long bones [10]. The femur fractures represent the third location of fractures in children [11]. The femur diaphysis fractures are frequent in males according to age, 11% affects children under two years of age, 21% between 3 and 5 years, 33% between 6 and 12 years, and 35% between 13 and 18 years [12,13]. The fracture may be incomplete, manifested by a transverse radiolucent line in the lateral cortex [14]. Some cases related to atypical femur fractures reported that the subtrochanteric area corresponds mainly to the spirals or longitudinal and transverse fractures [15].

The 3D modeling of bone tissue, in conjunction with the finite element method, has become a valuable tool in the orthopedic area since it helps to: determine the structural composition of the bone, predict the behavior of the bone subjected to various external agents, and design and optimize surgical implants [16,17,18]. Bone fractures can be simulated using tomography studies imported to software that computes the finite element method FEA, and they can differ from each other depending on the methodology used to carry out bone characterization and analysis [19]. The mineral density (BMD), in conjunction with finite element models, shows the possible fracture indifferent bone structures [20,21,22]. From another perspective, flexion and torsion tests were performed on critical areas subject to bone fracture OI [23,24,25]. Moreover, a numerical fracture report has been documented in 3D bone models, and various stress analyses on these models have been performed [26,27]. Additionally, other authors have performed numerical simulations to determine the behavior of the femur affected with OI considering particular case studies [28,29,30,31,32,33].

On the other hand, the OI patients are difficult to move in their normal environment, causing their parents to use temporary systems to transport them in vehicles safely. Unfortunately, these systems do not have a good design to protect children’s integrity with OI efficiently. In order to evaluate prototypes of vehicle transporting devices, some researchers have tried to reproduce the biomechanical conditions of children with OI into dummies [34,35,36]. Implementing this kind of device necessarily needs the biomechanical properties of bones of patients with OI to increase the bio-fidelity of the dummies [37]. This eventually will be useful in the design and development of future endoprosthesis that allows for the rehabilitation of bone damage, improving the quality of children’s lives with OI [38].

This work presents a 3D model of cortical and trabecular femur tissue affected by osteogenesis imperfecta disease type III. The femur was reproduced in a virtual environment using tomography images obtained from an OI patient of a three-year-old infant under the consented and informed agreement of their parents. Besides this, the biomechanical properties of the bone were determined by using Hounsfield units (HU) obtained directly from the tomographic images. Once the femur was modeled, the femur was evaluated in three ways corresponding to the most common types of fractures reported in the medical and scientific literature: transverse, oblique, and comminute. The payload and boundary conditions reproduced are computed by considering the weight of the patient. Finally, the incidences of fracture were compared through numerical analysis of the femur.

## 2. Materials and Methods

### 2.1. Methodology of 3D Modelling of an Infant’s Femur with OI Type III

From a tomographic study (CT) of a three-year-old patient affected with type III OI, Digital Imaging and Communication On Medicine (DICOM^®^, National Electrical Manufacturers Association, Arlington, VA 22209, USA) files were obtained that provided 1569 axial slices, which were manipulated in a 3D Image Segmentation and Processing Software (Scan IP^®^ program) to generate a virtual model femur. Initially, the area of the tomography was delimited to a section corresponding to the femur. Next, two layers were created to build the virtual model of the cortical and trabecular tissue that compose the femur. Subsequently, files with the *.stl extension were imported at the PowerShape^®^-e Student Edition program, which was used to create a mesh of surfaces that was part of a 3D solid model of the femur. Finally, files with the *.parasolid extension were generated, which can be identified by programs, such as Computer-aided design (CAD), that use the finite element method. Figure 1 shows the three-dimensional solid models corresponding to the cortical and trabecular tissue affected with OI.

### 2.2. Selection of the Minimum Unit of a 3D Image (Voxel)

In the ScanIP^®^ program, several voxels were delimited in the tomography for both cortical and trabecular tissue that compose the bone structure of the femur. Figure 2 shows the delimitation of the minimum unit of a tomography (voxel), based on the bone model of an infant with OI.

### 2.3. Hounsfield Unit (HU) Assessment and Apparent Density (ρ) Evaluations for OI Bone Type III

The Hounsfield scale has been universally adopted, assigning zero (0) to water and −1000 to air [39]. Hounsfield units and density have a linear correspondence, known as a function or calibration curve. The bone properties are modeled as a function of the apparent density, defined by mineralized mass divided by the total volume, including pores [40]. For this work, the calibration curve correction was performed using the methodology proposed by Taylor [41]. In this way, it is considered that the minimum apparent density of 0 g/cm^3^ corresponds to the minimum density of the trabecular tissue, and the maximum apparent density of 2 g/cm^3^ is associated with the maximum density of the cortical tissue. Fifty tomographic sections have been used to find the Hounsfield units along the femur bone, as shown in Figure 3, concentrating the highest values of HU in the central zone of the periphery of the femur diaphysis.

Figure 4 shows the graph of HU values obtained randomly for the cortical and trabecular tissue in each of the analyzed sections. Thus, the minimum value of −96 HU was determined in the trabecular tissue, and the maximum value was 962 HU for the cortical tissue. Thus, the relationship between apparent density and HU was finally established by Equation (1).
(1)$$\rho_{ap}=\frac{2}{1058}\mathrm{HU}+\frac{192}{1058},$$

### 2.4. Apparent Density (ρ) Evaluations for OI Bone Type III

The bone elastic modulus is defined transversely or longitudinally, according to the direction in which a force is applied. According to Wirtz [42], the necessary equations are reported to obtain the bone elastic modulus (E) in transversal and longitudinal form by using the bone density for the cortical and trabecular tissue. Moreover, the equations that define the Poisson’s coefficient (*ρ*) and the bone’s shear modulus (G) are reported. The OI bone’s biomechanical properties were computed using the apparent density values shown in Figure 3 and the Wirtz equations. Figure 5 shows the corresponding values to the relationship between the apparent density of the cortical tissue and Young’s Modulus in the longitudinal and transverse direction of the femur. The relationship shown in Figure 6 concerns trabecular tissue.

The Poisson’s ratio related to the bone tissue’s apparent density affected by OI was also determined. Since this constant considers the relationship between linear and transverse deformation, the same coefficient is considered longitudinal and transverse. In the graphs of Figure 7, the values obtained for the cortical and trabecular tissue can be observed.

The shear modulus of the cortical and cancellous tissues in the longitudinal and transverse direction was determined by the apparent density, Young’s modulus, and Poisson’s ratio. Figure 8 and Figure 9 show the values of the shear modulus for the apparent density.

### 2.5. Numerical 3D Model Analysis of OI Femur

The *.parasolid files were imported into ANSYS^®^ software (Canonsburg, PA, USA) to perform an orthotropic numerical evaluation of a femur with OI. The values determined are used to set up the numerical analysis of the femur shown in Table 1.

A 3D solid element of 20 nodes (SOLID186) was used to perform the mesh discretization since this element is compatible with contact elements such as TARGE170 and CONTA174. In the numerical evaluation, both elements were used to reproduce the interaction between the tissues. In general, the discretized mesh was carried out freely. According to this configuration, in total, 212,921 nodes and 142,018 elements were generated. The boundary conditions consider movement restriction and all the X, Y and Z axes’ rotation on the femur distal metaphysis, as shown in Figure 10 (magenta zone).

The Pauwels distributed load model was used to perform a femur analysis in the frontal plane, as a hip stabilization system in the monopodial support, a more critical condition in the gait cycle. The force exerted by the abductor’s muscles in this phase will be four times the infant’s weight. Considering a child with an OI weight of around 11 kg, a force equivalent of 431.64 N is obtained. This force was applied to the upper part of the femur head, as shown in Figure 11 (magenta zone) [43].

Finally, the birth and death function in software ANSYS© (Canonsburg, PA, USA) was used to create transverse, comminuted, and oblique fracture types on the model.

## 3. Results

Nominal stress resulting from the application of the von Mises theory is shown below. In each case, the maximum and minimum values in each analysis are indicated. In addition, the anterior and posterior of the femur model view is shown concerning the coronal anatomical axis.

### 3.1. Transverse Fracture Study

Figure 12a shows the transverse fracture in the medical image (radiography), and Figure 11 shows the finite element model where the fracture is generated and its appearance once the numerical analysis has been carried out.

Figure 13 shows the cuts of the model in each axis, showing the internal effect of the fracture intentionally generated in the femur bone, resulting in a high concentration of stress in the center of the femur shaft (the red areas), as this poses a significant risk when the patient’s femur is subjected to critical load distribution.

### 3.2. Oblique Fracture Study

The oblique fracture in a medical image (radiography) and the finite element model fracture are generated, and their appearance after the numerical analysis is shown in Figure 14.

Figure 15 shows the graphical and numerical result of the analysis carried out, where it can observe, in the first instance, the cuts of the model in each axis, showing the internal effect of the fracture intentionally generated in the femur bone, resulting in more areas with a high concentration of stress along the femoral shaft (the red areas).

### 3.3. Comminuted Fracture Study

Comminuted fracture in a medical image (radiography), the finite element model where the fracture is generated and its appearance once the numerical analysis has been carried out, is shown in Figure 16.

Figure 17 shows the graphical and numerical result of the analysis carried out, where we can observe in the first instance, the cuts of the model in each axis, showing the internal effect of the fracture intentionally generated in the femur bone, resulting in fewer areas of high-stress concentration in the upper part of the femur diaphysis (the red areas). However, the stress generated in that area is more significant than in the other analyses.

This section may be divided into subheadings. It should provide a concise and precise description of the experimental results, their interpretation, and the experimental conclusions that can be drawn.

## 4. Discussion

With the development of software tools and computer hardware, the validation of bone structure models has had a great boom, allowing for a trend towards personalized finite element modeling for each patient [44]. This facilitates the study of the behavior of bones affected with OI. Although the clinical characteristics in patients affected with this disease are general, each patient’s degree of bone deformation is a particular characteristic. Furthermore, the technological advancements in the area of tomography have allowed for the generation of models more attached to reality since the cortical and trabecular bone tissue can be segmented considering the grayscale, as mentioned by [33], or it can be considered by the values of the Hounsfield units, as shown in Section 2.3, and the models are shown in Figure 1. Furthermore, the finite element analysis can be performed considering both tissues, as shown by [31]. Moreover, other authors have studied femurs affected with OI from various criteria, for example, the degree of bone deformity [28], angular variation of the load application [29], increasing the load value [25,31], and influencing fracture [27]. A different method to reconstruction by tomography is shown, based on a statistical model of the shape and the appearance (SSAM) and a DXA image of the femur [26]. The use of tomography is not frequent in the diagnosis and follow-up for patients with osteoporosis. If a patient with OI has these studies, the analysis could be performed in this way. Finally, another author carries out a finite element study focused on the mechanical characterization of bone tissue affected with OI [23].

The mechanical properties of materials are an essential aspect in order to characterize them, determine their behavior, and predict under what circumstances a structural failure may present that compromises other entities within a system [24]. Derived from the above, a complete characterization of the bone tissue will define the weakest areas, where a fissure or fracture could occur under certain load conditions. Table 2 shows the values of Young’s Modulus, which is a parameter that characterizes the behavior of bone with OI, obtained by different authors. Since the bone tissue affected with OI is highly heterogeneous, predominantly trabecular tissue, defining it as a fragile material with little elasticity and the ability to support expected loads, all the different authors’ values are not uniform. Because performing physical tests on in vivo samples has a high cost and is difficult to reproduce, the non-invasive methodology described in this work is proposed. Indeed, the equations were described for bone tissue without any particular pathology. However, in the same way, they show that it is viable to develop this method to characterize bone tissue from a tomography performed on the limb to be analyzed, with the advantage that it can be easily reproducible and at almost zero cost compared to physical methods.

The curved anatomy of the bone is a favorable condition for crack propagation. The compression and tension stress, specifically the yellow- or orange-color zones shown in Figure 13, Figure 15 and Figure 17, points out areas more susceptible to fracture, and the red color indicates the most critical zone. Anisotropic analysis was carried out for the same previous study cases, taking the properties of the bone tissue affected with OI obtained by Fan (2006). The analyses were configured with the same boundary conditions established previously. The results are shown in Figure 18, Figure 19 and Figure 20 for the case of transverse, oblique, and comminuted fracture, respectively.

The isotropic analyses were compared to orthotropic analyses in the cases of transverse and oblique fracture, the stress values are remarkably similar, and the areas susceptible to fracture remain present. In contrast, in the case of comminuted fracture, the stress values and the behavior of the fabric are different.

## 5. Conclusions

The importance of the method developed in this work is the facility to build biological 3D models using the DICOM^®^ files. Likewise, this method is non-invasive because of the technology of the gadget used to obtain medical images. It can infer that the correct use of medical images due to the correct software can be helpful to determine the bone fracture behavior for certain pathologies, such as osteogenesis imperfecta, requites, osteopenia, and osteoporosis.

In most cases, the diagnostic is challenging before breaking limbs occurs. This alternative method by 3D bone models is different from the clinic method, preventing possible bone failure and fractures. For that reason, the exact image process can be helpful for OI patients. It can be perceived that the numerical analysis corresponding to the comminuted presents more significant stresses shown by the results exposed in the transverse and oblique fracture patterns; the area susceptible to failure extends in a wide radius around the crack where the fracture occurs. The transverse fracture presents minor stresses; however, it exhibits the same pattern of areas susceptible to bone failure, as in the case of the oblique fracture.

On the other hand, the results indicate that the transverse fracture presents the least stress and exposes lower critical areas. The values obtained in the different studies depend on the model achieved from the specific patient, that is, the type of OI, the patient’s age, and bone quality are essential factors, and the mechanical properties, the direction of the load, and the payload are significant in the results obtained. However, the areas susceptible to fracture are similar in most studies.

This study determined how the bone type of fracture affects its integrity to a lesser or greater degree. Furthermore, several complications are reported during the healing and bone regeneration after being exposed to an osteotomy. Therefore, it is recommended that the manner in which fractures are generated is considered an essential factor in how the tissue is regenerated [5,7]. On the other hand, X-ray images of the intramedullary canal determined the adequate diameter of the intramedullary implant for patients with OI, and the implant optimization is carried out since the finite element analysis is focused on the personalized study of each patient, which is considered critical for this type of syndrome.

Lastly, FEA predicts bone remodeling, which is essential for implementing bisphosphonates in patients to consolidate fractures. Finally, determining the areas susceptible to fracture in bones affected with OI is essential to redesign implants to avoid proximal fractures and unanchored implants in the distal areas of the bone.

## Figures and Tables

**Figure 1 children-08-01177-f001:**
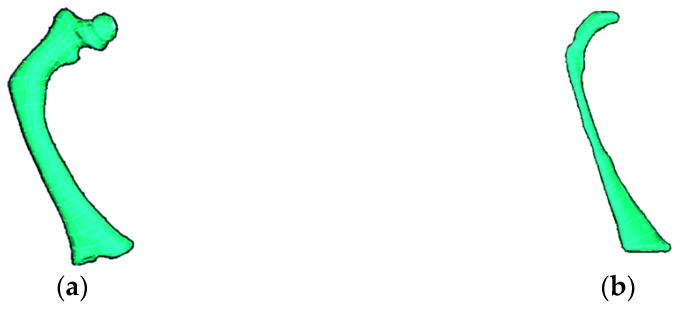
Models of cortical and trabecular tissue of the femur of an infant with type III OI: (**a**) cortical tissue; (**b**) trabecular tissue.

**Figure 2 children-08-01177-f002:**
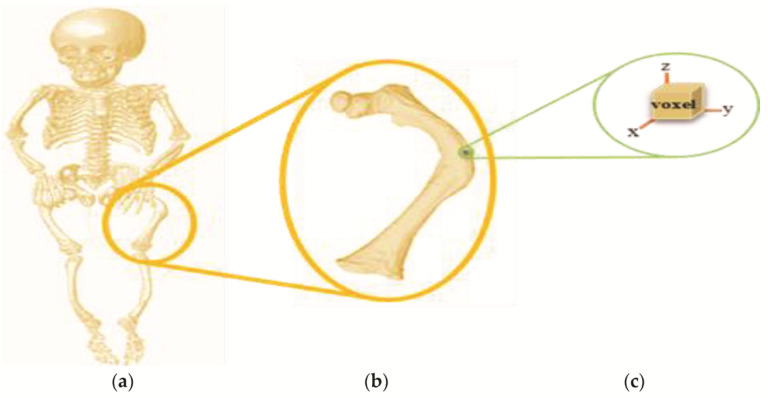
Determination of a voxel: (**a**) processing of all DICOM^®^ files; (**b**) selection of the tomographic slices corresponding to the femur; (**c**) delimitation of the minimum unit of a tomography.

**Figure 3 children-08-01177-f003:**
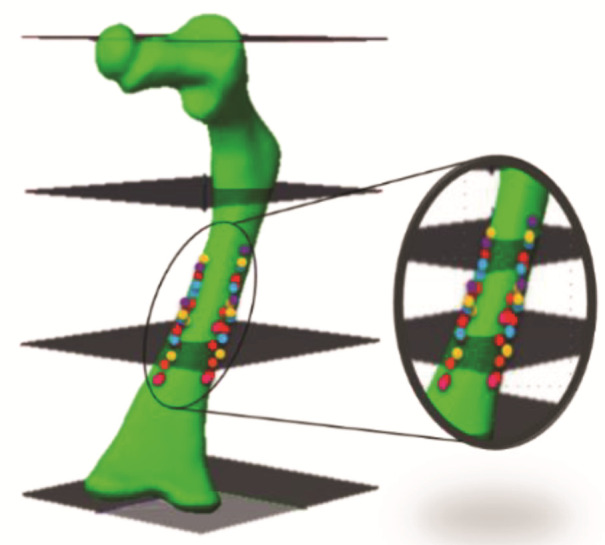
Obtaining HU values in different slices and areas.

**Figure 4 children-08-01177-f004:**
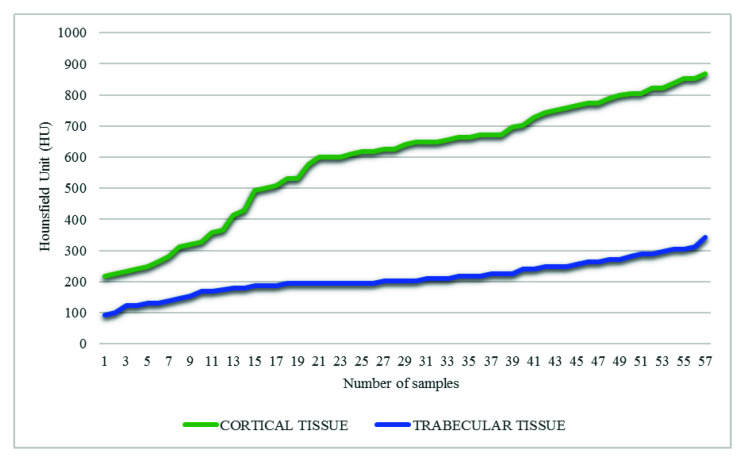
Values (HU) along the tomography of an affected femur with OI.

**Figure 5 children-08-01177-f005:**
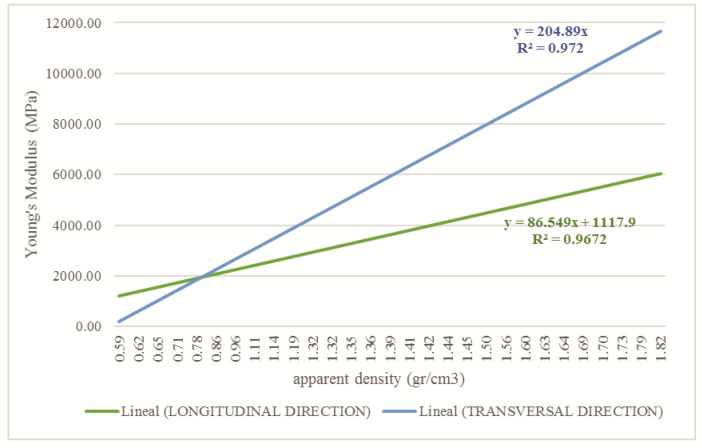
Relationship between apparent density and Young’s modulus for cortical tissue affected with OI in the longitudinal and transversal direction.

**Figure 6 children-08-01177-f006:**
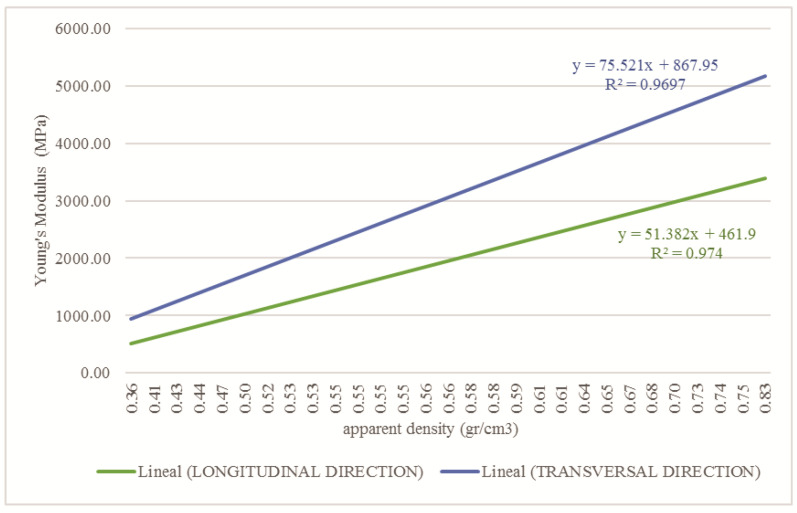
Relationship between apparent density and Young’s modulus for trabecular tissue affected with OI in the longitudinal and transversal direction.

**Figure 7 children-08-01177-f007:**
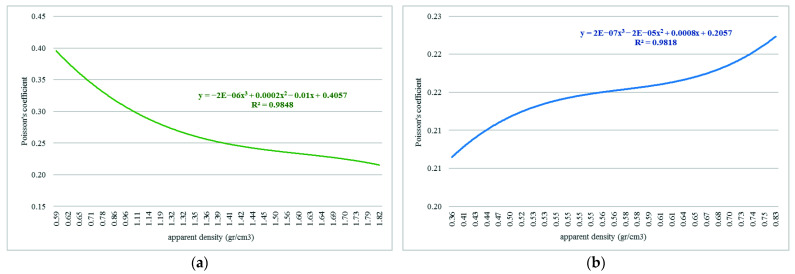
Relationship between apparent density and Poisson’s coefficient for bone tissue affected with OI: (**a**) cortical tissue; (**b**) trabecular tissue.

**Figure 8 children-08-01177-f008:**
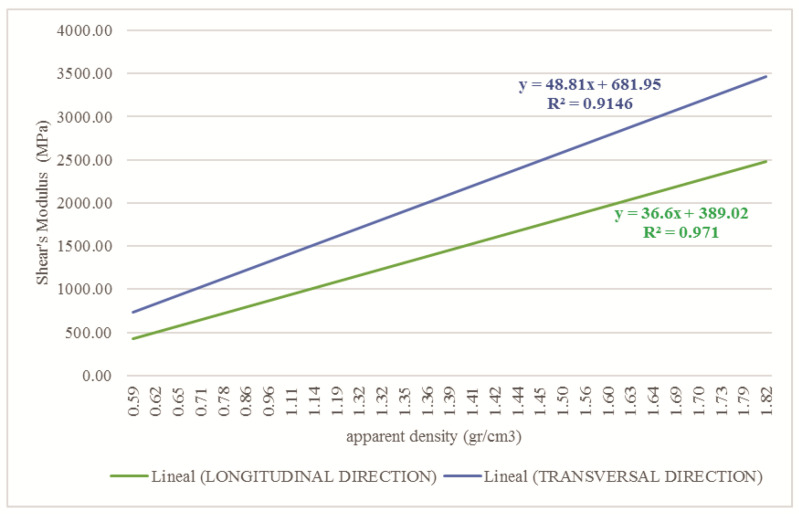
Relationship between apparent density and Shear’s modulus for cortical tissue affected with OI in the longitudinal and transversal direction.

**Figure 9 children-08-01177-f009:**
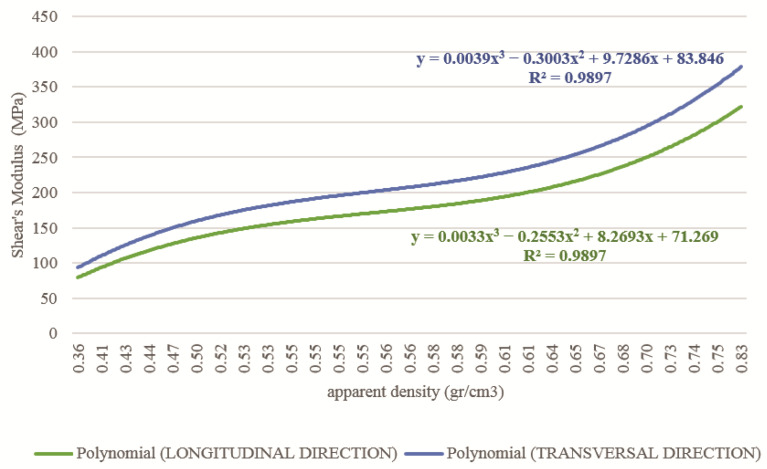
Relationship between apparent density and Shear’s modulus for trabecular tissue affected with OI in the longitudinal and transversal direction.

**Figure 10 children-08-01177-f010:**
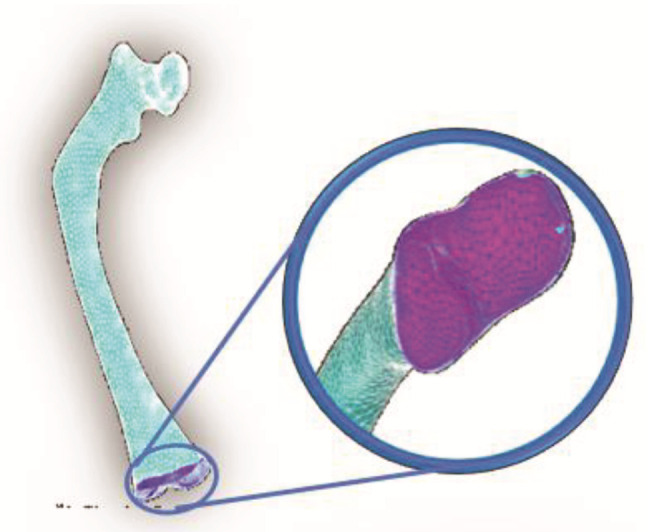
Border condition on the femur distal epiphysis, restricting all degrees of freedom.

**Figure 11 children-08-01177-f011:**
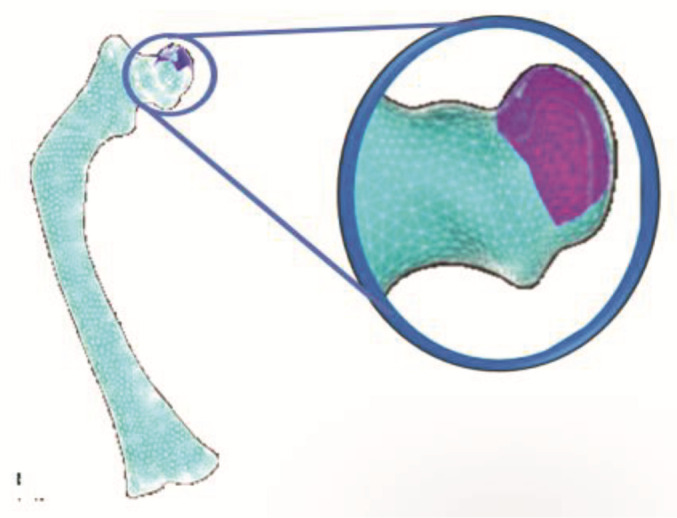
Force applied to the femur proximal epiphysis with a value of 431.64 N.

**Figure 12 children-08-01177-f012:**
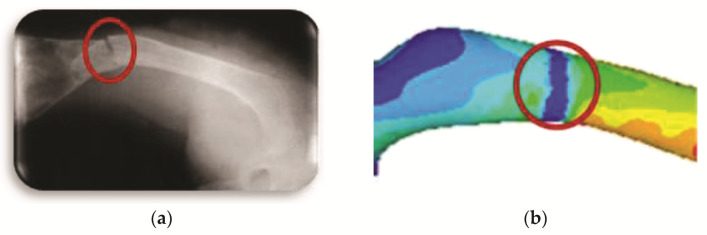
Transverse fracture: (**a**) medical image of bone fracture; (**b**) fracture of finite elements generated on the model.

**Figure 13 children-08-01177-f013:**
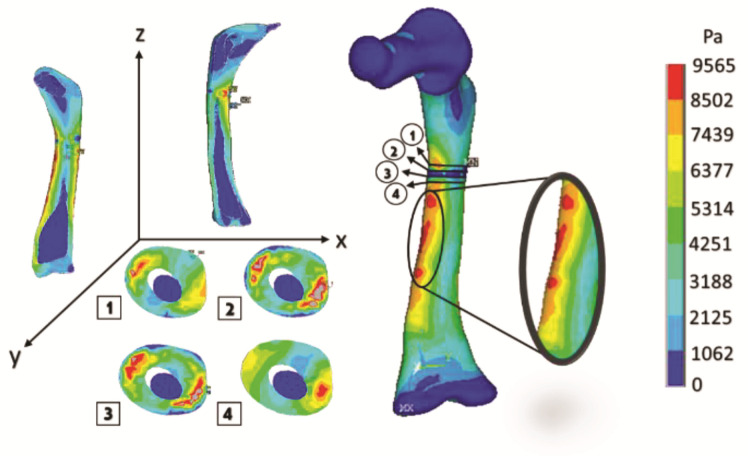
Orthotropic analysis. Von Mises stress showing maximum stress of 9.565 Pa.

**Figure 14 children-08-01177-f014:**
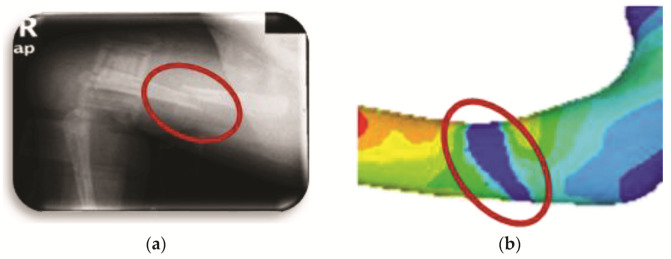
Oblique fracture: (**a**) medical image of bone fracture; (**b**) fracture of finite elements generated on the model.

**Figure 15 children-08-01177-f015:**
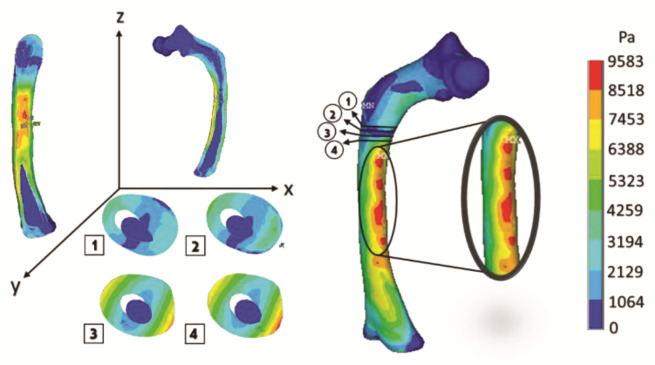
Orthotropic analysis. Von Mises stress showing maximum stress of 9.583 Pa.

**Figure 16 children-08-01177-f016:**
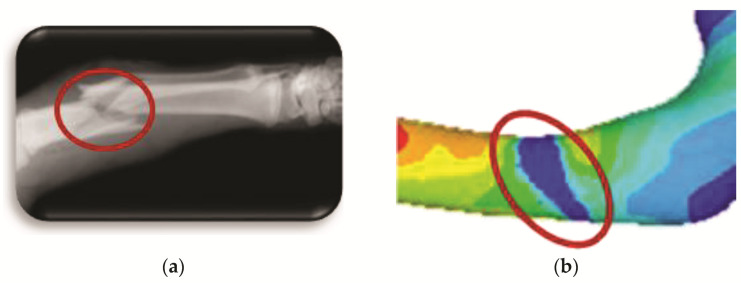
Comminuted fracture: (**a**) medical image of bone fracture; (**b**) fracture of finite elements generated on the model.

**Figure 17 children-08-01177-f017:**
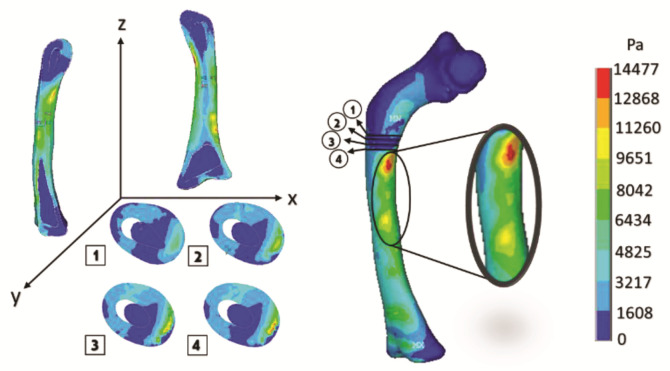
Orthotropic analysis. Von Mises stress showing maximum stress of 9.583 Pa.

**Figure 18 children-08-01177-f018:**
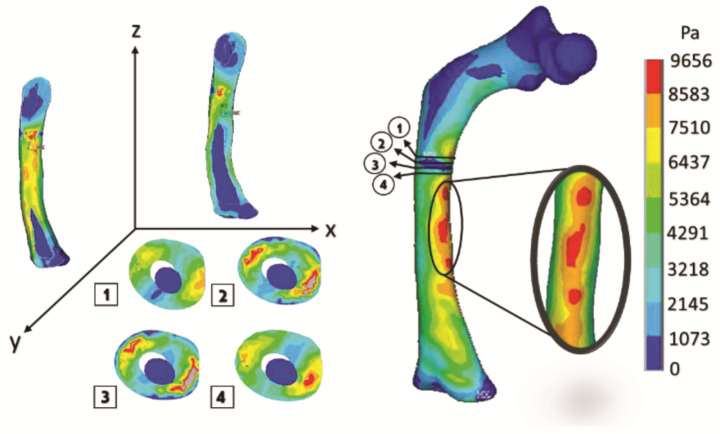
Isotropic analysis of the transverse fracture case. Von Mises stress showing maximum stress of 9.656 Pa.

**Figure 19 children-08-01177-f019:**
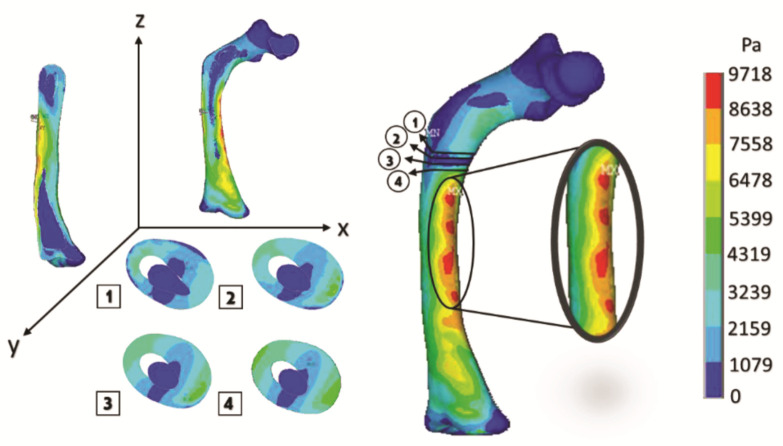
Isotropic analysis of the oblique fracture case. Von Mises stress showing maximum stress of 9.718 Pa.

**Figure 20 children-08-01177-f020:**
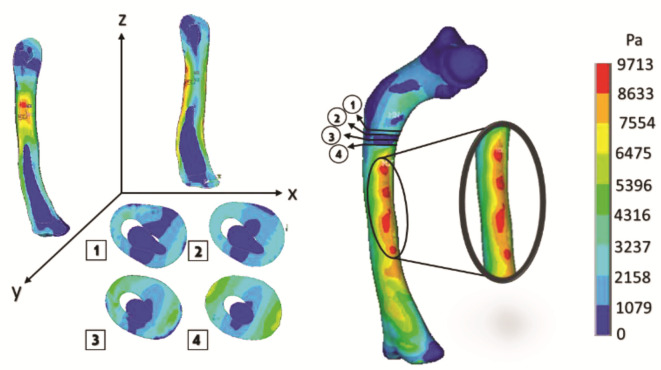
Isotropic analysis of the comminuted fracture case. Von Mises stress showing maximum stress of 9.713 Pa.

**Table 1 children-08-01177-t001:** Bone tissue with OI properties for the cortical and trabecular.

Mechanical Property	Tissue	
	Cortical	Cortical
Modulus of elasticity transversal $\left(E_{t}\right)$	5764.54 MPa	793.88 MPa
Modulus of elasticity longitudinal $\left(E_{p}\right)$	3627.78 MPa	449.41 MPa
Shear modulus $\left(G_{t}\right)$	2097.44 MPa	217.40 MPa
Shear modulus $\left(G_{p}\right)$	1450.43 MPa	184.79 MPa
Poisson’s ratio $\left(\upsilon_{pt}\right)$	0.27	0.21

**Table 2 children-08-01177-t002:** Mechanical properties of the bone tissue with OI for the cortical and trabecular.

Author	Year	Type OI	Method of Obtaining	Cortical E (GPa)	Trabecular E (GPa)
Fan et al. [45]	2006	III	Nanoindentation	15.22 (L) 13.92 (T)	13.60
Weber et al. [46]	2006	III–IV	Nanoindentation	21.3	
Fan et al. [47]	2007	III	Nanoindentation	19.19	18.56
Fan et al. [48]	2007	III	Nanoindentation	19.67	19.23
Albert et al. [49]	2013	III	Nanoindentation	16.3	
Albert et al. [50]	2014	IV	Mechanical tests	4.4 (L) 1.6 (T)	
Imbert et al. [51]	2014	−	Nanoindentation	17.6	
Vardakastani et al. [52]	2014	−	Mechanical tests	6.8	
Imbert et al. [53]	2015	−	Mechanical tests	4.0	

T, transversal. L, longitudinal. E, Young’s modulus.

## Data Availability

Not applicable.

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
