# Peer review of "Bone Fractures Numerical Analysis in a Femur Affected by Osteogenesis Imperfecta"

_children, 2021, doi:10.3390/children8121177_

Round 1

Reviewer 1 Report

3D modeling & FEA used in bone research is not new or emerging. It has been more than 30 years and helped us with better understanding of the complex behavior of bone. But why author use his or her method for 3D modeling & numerical analysis? What's the advantage (or disadvantage) compared to other methods when doing 3D modeling & analysis for Osteogenesis Imperfecta Disease & incidence of fracture in this disease? This question & hypothesis should be addressed, other than the study was just reporting feasibility of author's new method.

In Introduction, readers would like to know more about the gap of knowledge, such as any previous study about 3D modeling of femur affected by bone disease including Osteogenesis Imperfecta Disease; any previous study about numerical analysis to determine incidence of fracture; and any published study about other method than FEA to do 3D modeling.

Author did a great job in methods and materials. Results are credible. But it will be more instructive and attractive to readers (both clinicians and bone researchers) if authors compares his or her 3D modeling & numerical analysis with other methods, and even found some advantages over others. if author just show they can do 3D modeling & analysis, but, cannot show their 'product' is better than 'existing ones in the market', the study will not make that much sense

Author Response

REPLY TO REVIEWER 1

Ref. No.: 1435834

Title: Bone fractures numerical analysis in a femur affected by Osteogenesis Imperfecta

Journal: Children

Answer date: Nov. 30th, 2021

Corresponding author: Torres San Miguel C.R.

We thank the reviewers for their valuable comments.

Below are the responses to the reviewers' comments concerning the comments posted.

  1. 3D modeling & FEA used in bone research is not new or emerging. It has been more than 30 years and helped us with better understanding of the complex behavior of bone.But why author use his or her method for 3D modeling & numerical analysis? What's the advantage (or disadvantage) compared to other methods when doing 3D modeling & analisys for Osteogenesis Imperfecta Disease & incidence of fractura in this disease? This question & hypothesis should be addressed, other than the study was just reporting feasibility of author's new method.

Ok, thanks for the observation. As you well mentioned, the MEF is a tool that has allowed us to understand and determine the behavior of bones. Most studies focus on the characterization of healthy bone tissue since OI is a disease with low incidence, they have limited. The works that focus on the behavior of this disease and try to generalize it are already complicated. It is called "The Syndrome of Osteogenesis Imperfecta" due to the multiple genetic variations and specific clinical characteristics of each type of OI, the methodology proposed from Computerized axial tomography has also been used previously. However, it is limited in bones affected with OI. The study of the behavior of these bones, mainly determining the areas susceptible to fracture, has been useful in the redesign of intramedullary implants, specifically the redesign of a telescopic nail for patients affected with OI [7]. In addition, these results can serve as a reference for orthopedists since more or more minor bone damage can be observed depending on the type of fracture performed in an osteotomy.

  1. In Introduction, readers would like to know more about the gap of knowledge, such as any previous study about 3D modeling of femur affected by bone disease including Osteogenesis Imperfecta Disease, any previous study about numerical analysis to determine incidence of fracture; and any published study about other method than FEA to do 3D modeling.

Paragraph 5 mentions some works referring to studies carried out by other researchers on the methodologies used to characterize and analyze bones with Osteogenesis Imperfecta and the possible areas of fractures generated in bone tissue affected with OI according to various external considerations. However, we have added new references to improve the paper.

  1. Author did a great job in methods and materials. Results are credible. But it will be more instructive and attractive to readers (both clinicians and bone researchers) if authors compares his o her 3D modeling & numerical analysis with other methods, and even found some advantages over others. If author just show they can do 3D modeling & analysis, but, cannot show their product is better than existing ones in the market, the study will not make that much sense.

The explanation of discussions and conclusion sections have been modified to match this observation. Thank you in advance for your comments. We believe they were very assertive and improved the work presented.

Reviewer 2 Report

It's a very interesting paper regarding a matter that I hope it will be improved in the future in order to prevent fractures in these particolar kind of diseases. Moreover, this kind of method could be useful to design custom made devices to synthesis the fractures and their positioning on the bone.

I demand only a slightly improvement in the English production, maybe with a native language editor if possible. 

In particular,  in line 34 I will substitute the word "trouble" with "condition";

in line 35 "causing that their parents need to use" I will substitute with "causing their parents to use";

In caption for figure 20 you wrote "oblique" instead of "comminuted" I believe for a typo reading all the text. Please confirm it.

Author Response

REPLY TO REVIEWER 2

Ref. No.: 1435834

Title: Bone fractures numerical analysis in a femur affected by Osteogenesis Imperfecta

Journal: Children

Answer date: Nov. 30th, 2021

Corresponding author: Torres San Miguel C.R.

We thank the reviewers for their valuable comments.

Below are the responses to the reviewers' comments concerning the comments posted.

It´s a very interesting paper regarding a matter that I hope it will be improved in the future in order to prevent fractures in these particular kind of diseases. Moreover, this kind of method could be useful to design custom made devices to synthesis the fractures and their positioning on the bone.

  1. I demand only a slightly improvement in the English production, maybe with a native language editor if possible.

Ok, thank you for the observation. We will review it.

  1. In particular, in line 34 I will substitute the word “trouble” with “condition”.

Done.

  1. In line 35 “causing that their parents need to use” I will substitute with “causing their parents to use”

Done.

  1. In caption for figure 20 you wrote “oblique” instead of “comminuted” I believe for a typo reading all the text. Please confirm it.

Done.

Round 2

Reviewer 1 Report

All comments are addressed